# Flash glucose monitoring with the FreeStyle Libre 2 compared with self-monitoring of blood glucose in suboptimally controlled type 1 diabetes: the FLASH-UK randomised controlled trial protocol

Emma G Wilmot [1,2] Mark Evans,[3] Katharine Barnard-Kelly [4] M Burns,[5] Iain Cranston,[6] Rachel Ann Elliott [7] G Gkountouras,[7] N Kanumilli,[8,9] A Krishan,[10] C Kotonya,[1] S Lumley,[11] P Narendran,[12] Sankalpa Neupane,[13] Gerry Rayman [14] Christopher Sutton,[10] V P Taxiarchi,[10] H Thabit,[9,15] L Leelarathna[9,15]

► Prepublication history and additional online supplemental material for this paper are available online. To view these files, please visit the journal online (http://dx.doi.org/10.1136/bmjopen-2021-050713).

For numbered affiliations see end of article.

**Correspondence to**
Dr L Leelarathna;
Lalantha.Leelarathna@mft.nhs.uk

## ABSTRACT

**Introduction** Optimising glycaemic control in type 1 diabetes (T1D) remains challenging. Flash glucose monitoring with FreeStyle Libre 2 (FSL2) is a novel alternative to the current standard of care self-monitoring of blood glucose (SMBG). No randomised controlled trials to date have explored the potential benefits of FSL2 in T1D. We aim to assess the impact of FSL2 in people with suboptimal glycaemic control T1D in comparison with SMBG.

**Methods** This open-label, multicentre, randomised (via stochastic minimisation), parallel design study conducted at eight UK secondary and primary care centres will aim to recruit 180 people age ≥16 years with T1D for >1 year and glycated haemoglobin (HbA1c) 7.5%–11%. Eligible participants will be randomised to 24 weeks of FSL2 (intervention) or SMBG (control) periods, after 2-week of blinded sensor wear. Participants will be assessed virtually or in-person owing to the COVID-19 pandemic. HbA1c will be measured at baseline, 12 and 24 weeks (primary outcome). Participants will be contacted at 4 and 12 weeks for glucose optimisation. Control participants will wear a blinded sensor during the last 2 weeks. Psychosocial outcomes will be measured at baseline and 24 weeks. Secondary outcomes include sensor-based metrics, insulin doses, adverse events and self-report psychosocial measures. Utility, acceptability, expectations and experience of using FSL2 will be explored. Data on health service resource utilisation will be collected.

**Analysis** Efficacy analyses will follow intention-to-treat principle. Outcomes will be analysed using analysis of covariance, adjusted for the baseline value of the corresponding outcome, minimisation factors and other known prognostic factors. Both within-trial and life-time economic evaluations, informed by modelling from the perspective of the National Health Service setting, will be performed.

**Ethics** The study was approved by Greater Manchester West Research Ethics Committee (reference 19/NW/0081). Informed consent will be sought from all participants.

**Trial registration number** NCT03815006.

## Strengths and limitations of this study

► Flash-UK is a multicentre randomised controlled trial of the novel FreeStyle Libre 2 flash glucose monitor over a 6-month follow-up period assessing the impact on people living with type 1 diabetes and suboptimal glucose control, in comparison to self-monitoring of blood glucose, with a primary outcome of glycated haemoglobin.

► It is the first randomised study to assess the clinical efficacy and health economic benefits of the Libre 2 device providing high quality data for UK policy-makers.

► The integration of a virtual assessment pathway into the trial design ensures eligible candidates are not restricted from participation owing to the COVID-19 pandemic.

► A wide range of secondary outcomes including continuous glucose monitoring data and psychosocial outcomes will provide detailed insight into the impact of this technology on people living with type 1 diabetes.

► The study is open (unblinded) and conducted in UK National Health Service (NHS) only. Some findings from the study may only be applicable to UK NHS setting.

**Protocol version** 4.0 dated 29 June 2020.

## INTRODUCTION

Type 1 diabetes mellitus (T1D) is one of the most common endocrine conditions. It is estimated that approximately 415 million adults (5%–15% T1D) and 520 thousand children (95% T1D) worldwide suffer from diabetes.[1] Despite the availability of therapeutic options

such as self-monitoring of blood glucose (SMBG), structured education, rapid-acting insulin analogues and insulin pump therapy, glycaemic control in the majority of people with T1D remains suboptimal[2] and they therefore remain prone to complications associated with high glucose levels, such as kidney failure and blindness.[3] In England, less than one-third of people with T1D achieve a glycated haemoglobin (HbA1c) level <7.5%.[4] Studies have shown a strong relationship between the frequency of SMBG and HbA1c, with the National Institute of Clinical Excellence recommending 4–10 checks per day.[5 6] However, due to pain, inconvenience and the limited information a moment-in-time glucose value provides, finger-stick glucose monitoring remains a key barrier in achieving near normal glucose levels.

In 2014, the FreeStyle Libre Flash Glucose Monitoring System (FSL) (Abbott Diabetes Care, Oxon, UK) became available as a potential alternative to SMBG. The sensor utilises wired enzyme technology[7] to continuously measure interstitial glucose levels. The arm-worn sensor is scanned using a reader or mobile phone app and provides information on current and previous glucose levels and trends. The IMPACT randomised controlled multicentre European trial was the largest study to evaluate the FSL[8] in 328 participants with well-controlled (HbA1c ≤7.5%, 59 mmol/mol) T1D, one-third of whom used continuous subcutaneous insulin infusion (CSII) therapy. FSL use was associated with improvement in a range of glucose-related outcomes: a 38% reduction in time spent in hypoglycaemia (<3.9 mmol/L). There was an increase in glucose time in range but HbA1c was unchanged. The impact of FSL was also assessed in those with T2D on intensive insulin therapy in a large multicentre randomised European study of 224 participants.[9] Time in hypoglycaemia (<3.9 mmol/L) reduced compared with controls by 43% but HbA1c was unchanged. In both randomised controlled trials treatment satisfaction was higher in FSL users and no device-related serious adverse events were reported, suggesting that flash glucose monitoring also offers a safe replacement to SMBG in those with diabetes on intensive insulin therapy.

Subsequently a range of observational studies have demonstrated benefits of FSL use for HbA1c and hypoglycaemia.[10–15] Campbell et al evaluated the use of FSL as a replacement for SMBG in young people (4–17 years) (n=76, 58% CSII users, mean (SD) age 10.3 (4.0) years, baseline HbA1c 7.9 (1.0)% (63 mmol/mol), T1D duration 5.4 (3.7) years) with T1D in a single-arm European multicentre trial.[16] After 2 weeks' baseline masked wear, participants used FSL for 8 weeks. HbA1c significantly improved vs baseline, −0.4±0.6%, p<0.0001. However, a subsequent 6-month randomised controlled parallel-arm trial of the FSL in 64 participants aged 13–20 years with T1D and HbA1c ≥9% (≥75 mmol/mol) demonstrated no statistically significant difference between groups for changes in HbA1c at 6 months (adjusted mean −0.2% greater improvement for FSL [95% CI −0.9 to 0.5] (−2.1 mmol/mol (95% CI −9.6 to 5.4)); p=0.58).[17] The

discrepancy in findings between the observational and randomised controlled trials in the paediatric population highlights the importance of high-quality randomised trials to investigate the benefits of this technology. To date, although some are planned, there has been no published randomised controlled trial to demonstrate the impact of the FSL in adults with T1D and high HbA1c levels.[18 19]

More recently, in 2020, the FSL 2 (FSL2) launched in Europe. This is very similar to FSL except for an optional additional alarm to alert users when the glucose level falls outside their target range, in addition to improved senor accuracy.[20] The FSL2 was launched in the United Kingdom in January 2021. To the best of our knowledge, no randomised controlled trials to assess the efficacy of the FSL2 system has been conducted or in progress. No economic evaluation assessing relative costs and benefits has been carried out in this patient group, nor has there been an assessment of patient acceptability. The purpose of the Flash UK study is to address this gap in the evidence and determine whether use of flash glucose monitoring with the FSL2 device will improve HbA1c over a 24-week randomised period compared with SMBG in adults with T1D and suboptimal glycaemic control.

## METHODS AND ANALYSIS
### Trial design
Flash-UK is an open, multicentre, randomised (1:1), parallel-group superiority trial, in adults and adolescents (16 years and older) with T1D and suboptimal glycaemic control (HbA1c 7.5% to 11% (59 to 97 mmol/mol), either on CSII or multiple daily injections (MDI), contrasting flash glucose monitoring using the FSL2 device with traditional finger-stick SMBG for 24 weeks. The study flow chart is outlined in figure 1.

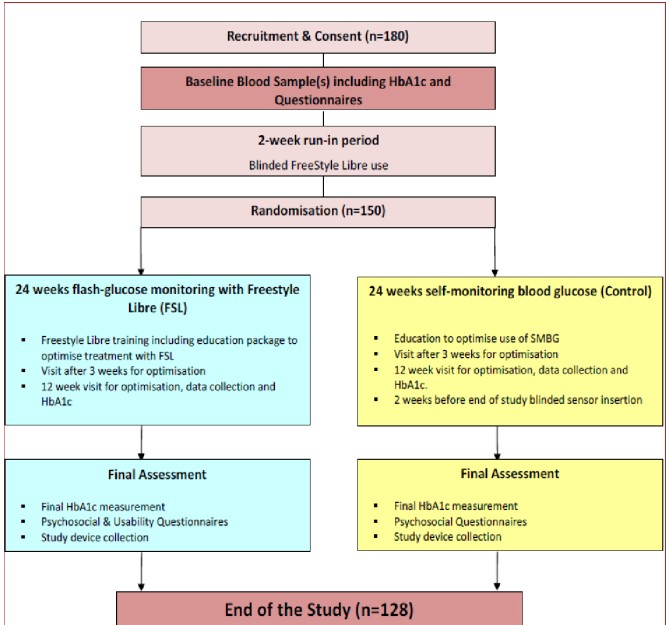

**Figure 1** Flash UK study flow chart. HbA1c, glycated haemoglobin; SMBG, self-monitoring of blood glucose.

## Study setting

Eight (one primary and seven secondary) care diabetes services from across England, UK.

1. Diabetes Centres within Manchester University Foundation Trust.
2. Diabetes Centres within University Hospitals of Derby and Burton National Health Service (NHS) Foundation Trust
3. University Hospitals Birmingham NHS Foundation Trust.
4. Wolfson Adult Diabetes Endocrine Clinic, Cambridge Universities Hospitals, Cambridge.
5. Norfolk and Norwich University Hospital, Norwich.
6. Queen Alexandra Hospital, Portsmouth.
7. Ipswich Hospital, East Suffolk and North Essex NHS Foundation Trust, Ipswich.
8. Wareham Surgery (Wareham) and The Adam Practice (Poole), NHS England Primary Care general practitioner Practices.

## Eligibility criteria

The major eligibility criteria are age ≥16 years with T1D for at least 1 year and a HbA1c between 7.5% and 11%. The full criteria are available in box 1.

## Interventions

### FreeStyle Libre 2

This intervention is the CE marked FSL2 flash glucose monitoring device (Abbott Diabetes Care, Oxon, UK). The FSL2 glucose sensor is an arm worn sensor intended to last for 14 days. The component not directly attached to the patient is the handheld reader and/or mobile phone app which displays current and historical glucose data. Education and training about insertion and initiation of the sensor as well as how to use flash-glucose monitoring data for treatment optimisation is also provided. Encouragement is also provided to download data at home to identify pattern recognition. This session, designed to meet the needs of the individual, is conducted by a professional diabetes educator or a member of the study team.

### Finger-stick SMBG

This is continuation of usual treatment. Additionally, encouragement will be given to use finger-stick glucose levels to optimise therapy and education about insulin dose adjustments using finger-stick glucose levels will also be provided. Participants in both arms will also receive training on sick day rules and dealing with hypo and hyperglycaemia as required.

Participants will be provided with an information leaflet following the training session, although the information provided will be tailored to the respective intervention. The leaflet provided in conjunction with the FSL2 will include sign-posting to educational videos provided by the Association of British Clinical Diabetologists (https://abcd.care/dtn/education) and Bertie online (www.bertieonline.org.uk) in conjunction with the finger-stick SMBG.

### Box 1 Key inclusion and exclusion criteria

**Inclusion criteria**

1. The participant is ≥16 years old.
2. The participant has type 1 diabetes, as defined by WHO for at least 1 year or is confirmed C-peptide negative if duration of diabetes is <1 years.
3. Participant is treated with insulin pump or multiple daily injection for at least 12 weeks and no plans to change treatment modality during next 28 weeks.
4. The participant is literate in English for safe study conduct.
5. Screening glycated haemoglobin ≥7.5% (58.5 mmol/mol) and ≤11% (97 mmol/mol) based on analysis from local, central or third party external laboratory.
6. The participant is willing to wear study glucose sensor and scan for glucose levels at regular intervals.
7. The participant is willing to follow study-specific instructions and improve glucose control.
8. Female participants of childbearing age should be on effective contraception or not sexually active/no plans for pregnancy.

**Exclusion criteria**

1. Non-type 1 diabetes mellitus including those secondary to chronic disease.
2. Any other physical disease or people with known severe mental illness (psychotic disorder, bipolar disorder, dementia, substance and alcohol dependence, learning disabilities, depression with active suicidal ideation) which are likely to interfere with the normal conduct of the study and interpretation of the study results as judged by the investigator.
3. Current users of real-time glucose monitoring sensors or flash-glucose monitoring for more than 4 weeks within last 12 weeks.
4. Initiation of medications/treatments known to interfere with glucose metabolism (eg, metformin, SGLT2 inhibitors, GLP-1 agonists, pramlinatide) within the last 6 weeks or planning to start these medications within the next 6 months (patients on stable treatment is not an exclusion) or current or planned glucocorticoid use other than inhaled/topical use.
5. Known or suspected allergy against insulin.
6. Severe visual impairment.
7. Complete loss of hypoglycaemia awareness.
8. Patient receiving dialysis/predialysis based on history.
9. More than one episode of severe hypoglycaemia as defined by American Diabetes Association[30] in preceding 24 weeks.
10. Pregnancy, planned pregnancy in the next 8 months or breast feeding.

## Outcomes

The FLASH-UK study is designed to assess an extensive array of clinical, psychosocial and usability outcomes (table 1). An outline of a dedicated health economic evaluation is provided below. The primary end point is HbA1c level at 24 weeks post randomisation. Sensor based outcomes comparing the FSL2 with SMBG will be assessed for the last 2 weeks of the intervention period. Prespecified sensor based outcomes include time in the target range (3.9–10 mmol/L), duration of hypoglycaemia <3.5 mmol/L (63 mg/dL); <3.0 mmol/L (54 mg/dL); <2.8 mmol/L (50 mg/dL), duration of hyperglycaemia (>10 mmol/L (180 mg/dL), and >16.7 mmol/L (>300 mg/dL) and glucose variability (SD and coefficient

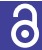

| Table 1 Secondary outcomes | | |
|---|---|---|
| **Category** | **Outcomes** | **Assessment time point** |
| HbA1c based | HbA1c<br>HbA1c ≤53 mmol/mol (7.0%) (yes/no)<br>HbA1c ≤59 mmol/mol (7.5%) (yes/no)<br>Reduction in HbA1c ≥5.5 mmol/mol (0.5%) from baseline (yes/no)<br>Reduction in HbA1c ≥11 mmol/mol (1.0%) from baseline (yes/no) | Baseline and 12 weeks<br>Baseline, 12 weeks and 24 weeks<br>Baseline, 12 weeks and 24 weeks<br>Baseline, 12 weeks and 24 weeks<br>Baseline, 12 weeks and 24 weeks |
| Sensor based (glucose) | Time spent in the target glucose range 3.9–10.0 mmol/L (70–180 mg/dL)<br>Time spent below target glucose (<3.9 mmol/L) (<70 mg/dL), <3.5 mmol/L (63 mg/dL), <3.0 mmol/L (54 mg/dL); <2.8 mmol/L (50 mg/dL)<br>Time spent above target glucose (10.0 mmol/L) (180 mg/dL), >16.7 mmol/L (300 mg/dL)<br>Average glucose, SD, coefficient of variation<br>AUC of glucose below 3.0 mmol/L (54 mg/dL) | Baseline and 24 weeks |
| Non-sensor based (clinical) | Daily average total insulin dose<br>Daily average basal insulin dose<br>Daily average bolus dose<br>Average number of boluses of rapid acting insulin per day<br>Frequency of severe hypoglycaemic episodes as defined by American Diabetes Association<br>Frequency of significant ketosis events (plasma ketones >3 mmol/L)<br>Nature and severity of other adverse events | Baseline, 12 weeks and 24 weeks |
| Psychosocial | Type 1 Diabetes Distress Scale<br>Quality of Life (EQ-5D-5L)<br>Patient Health Questionnaire<br>Diabetes fear of injecting and self-testing questionnaire<br>The revised Diabetes Eating Problem Survey | Baseline and 24 weeks |
| Process evaluation (utility and acceptability) | FSL2 device utilisation data, including: average no of scans per day (7:00–23:00 hours), per night (23:00–7:00 hours) and over the full 24-hour period; average no of days of usage per week for the full 24 weeks intervention | Continuous (FSL arm only) |
| | Number of finger-stick glucose level tests per day | Continuous |
| | Diabetes Treatment Satisfaction Questionnaire<br>Glucose Monitoring Satisfaction Survey | Baseline and 24 weeks |

AUC, area under the curve; FSL2, FreeStyle Libre 2; HbA1c, glycated haemoglobin.

of variation). All the sensor based metrics will also be analysed separately for daytime (7:00–23:00 hours) and night-time (23:00–7:00 hours) in addition to the 24-hour period. Insulin usage data will be compared between the two arms. Harms outcomes include the frequency of severe hypoglycaemic episodes as defined by American Diabetes Association, frequency of significant ketosis events (plasma ketones >3 mmol/L) and the nature and severity of other adverse events. Information on any other antidiabetes therapy will be collected.

Questionnaires (table 1) will be employed at baseline and the end of the study to evaluate participants' responses in terms of quality of life, diabetes distress, needle burden, disordered eating, depression and diabetes treatment satisfaction using EQ-5DL-5L questionnaire, Type 1 Diabetes Distress Scale, Diabetes Fear of Injecting and Self-testing questionnaire, Diabetes Eating Problem Survey, Diabetes Treatment Satisfaction Questionnaire, Patient Health Questionnaire-9 and The Glucose Monitoring Satisfaction Survey. Hypoglycaemia burden will be assessed using Clarke questionnaire and Gold score.

### Participant timeline and data collection

The study will consist of six visits for those in the FSL2 arm and seven visits in the SMBG arm. Visits 1 and 2 can be conjoined. The study flow chart is shown in figure 1. Key activities undertaken during each study visit are shown in table 2.

Due to the COVID-19 pandemic all study visits can be conducted either face to face or virtually (supported by telephone and videoconferencing as appropriate) as indicated in the approved study advertisement and participant information sheet. At visit 1, following informed written consent (model consent form and patient information sheet in online supplemental material) by trained

**Table 2** Schedule of study visits

| Visit/contact | Description | Time since randomisation | Start relative to previous/next visit/activity (+/- 2 weeks of planned visit date) |
|---|---|---|---|
| Visit 1 | Recruitment and Screening visit: Consent HbA1c, baseline bloods, baseline questionnaires | −2 to −3 weeks | – |
| Visit 2 | Blinded flash glucose monitor insertion | −2 weeks | Within 1 to 2 weeks of visit 1. Can coincide with visit 1 |
| Visit 3 | Adherence assessment &and Randomisation FSL2/Self- monitoring of glucose initiation ▶ Education | 0 weeks | After 2 weeks of visit 2 |
| Visit 4 | Review data/optimisation Collect participant diary | +4 weeks | After 4 weeks of visit 3 |
| Visit 5 | Review data/optimisation. Data download ▶ HbA1c ▶ Collect participant diary | +12 weeks | After 8 weeks of visit 5 |
| Visit 6 | Blinded flash glucose monitor insertion (extra visit in SMBG arm) | +22 weeks | After 10 weeks of visit 5 |
| Visit 7 | End of self-monitoring intervention arm ▶ HbA1c. ▶ Questionnaires ▶ Collect participant diary | +24 weeks | 2 weeks after visit 6 |

FSL2, FreeStyle Libre 2; HbA1c, glycated haemoglobin; SMBG, self-monitoring of blood glucose.

members of the research team, medical and diabetes history will be recorded including presence of diabetes complications, hypoglycaemia burden including the use of Clarke and Gold questionnaires, use of concomitant diabetes medications, ethnicity, body weight and height measurement; demographic data, insulin therapy, occupation and educational attainment, any history of disordered eating or needle phobia, previous participation in structured education, carbohydrate counting status, use of bolus calculator and patient self-report psychosocial questionnaires will be completed. Blood samples will be taken to measure HbA1c.

At visit 2, the FSL Pro blinded continuous glucose monitoring (CGM) device will be inserted, to be worn for 2 weeks. For participants on the virtual pathway, participant will be taught to self insert the sensor. At visit 3, participant adherence/tolerance of using the flash-CGM over the preceding 14 days will be assessed. To proceed to randomisation at least 10 of 14 days blinded CGM data must be available. Those who have <10 days data will be provided with a new FSL pro sensor and reader to obtain the minimum data requirements prior to randomisation or discontinue from trial participation if there is a continued adverse reaction to the sensor adhesive.

Participants will be followed up at 4, 12 and 24 weeks postrandomisation (±2 weeks). At these visits glucose data will be downloaded and participant diaries which collect information on insulin doses and carbohydrate intake in last 5 days will be collected. Participants in the virtual pathway will be encouraged to upload data from home. Blood tests for HbA1c will also be performed at 12 and 24

weeks after randomisation. Those in the SMBG group will attend an additional visit at 22 weeks to have the blinded CGM sensor fitted to be worn from 22 to 24 weeks. At 24 weeks, body weight measurement will be made where possible and the participant will be asked to complete questionnaires. In addition, participants in the FSL2 arm will be asked to complete a questionnaire exploring expectations and experience of using FSL2 during the study. Healthcare resource use in primary and secondary healthcare settings will also be collected at 24 weeks.

Adverse events will be routinely collated at study visits. The independent data monitoring committee (IDMC) will be informed of all serious adverse events and any unanticipated adverse device effects that occur during the study and will review compiled adverse event data at periodic intervals.

### Sample size
The target effect size (minimally clinically important HbA1c difference) of 0.4% was chosen as this is consistent with other relevant trials (0.5% in REPOSE,[21] 0.4% in DIAMOND,[22] 0.3% in GOLD[23]). To achieve 90% power using an independent-sample t-test for 24 week HbA1c values (2-tail alpha=0.05, power=0.80, effect size=0.4%, SD=0.8%,[23] ie, standardised effect size=0.5), 128 participants with primary outcome data are required. This is inflated to a target of 150 randomised (assuming maximum 15% attrition) and up to 180 recruited (to allow for prerandomisation losses). The use of analysis of covariance (ANCOVA) with adjustment for baseline

values of HbA1c and other minimisation factors should increase power.

## Recruitment

Participants will be identified by treating clinicians in each centre and participant information sheet provided. The first study site was activated on the 20 December 2019 and first participant was recruited on the 9 January 2020. Each site will each aim to recruit between 14 and 30 participants up to a total of 180 participants. The study has been advertised via social media and through the official trial website (https://sites.manchester.ac.uk/flash/), providing those who do not usually attend study centres to have the opportunity to participate. As of 31 January 2021, 109 participants had been randomised. It is anticipated that recruitment will be completed by 30 April 2021. In the event that a participant meets withdrawal criteria, sites will engage with participants to seek their verbal agreement to volunteer their continued participation for HbA1c time points (post randomisation) and exit questionnaires under the intention-to-treat (ITT) analysis principle.

## Allocation
### Sequence generation

Allocation to one of the two intervention arms (24 weeks use of flash glucose monitoring or 24 weeks use of conventional finger-stick SMBG) will use stochastic minimisation (factors: study centre, baseline HbA1c (7.5%–9.0%; >9.0%–11%), treatment modality (MDI; CSII), prior participation in structured education course (yes; no) and current use of bolus calculator (yes; no)).

### Allocation concealment mechanism and its implementation

Allocation will be implemented using the web-based Sealed Envelope software (Sealed Envelope, London, UK) randomisation system. This will be independently managed by Manchester Clinical Trials Unit (MCTU) staff. Participants will have their pseudonymised registration details entered into the randomisation system by delegated recruitment site research teams who will also confirm participant eligibility; only then will the system generate an email confirmation with their allocation.

## Blinding (masking)

The trial is fully open. It is not possible to blind investigators or participants to the delivery or receipt of the intervention; data collectors and statisticians are also unblinded to treatment allocations.

## Laboratory analysis

Blood samples for the measurement of HbA1c levels will be taken at three different time points: screening, 12 and 24 weeks. This can be completed using the local laboratory (face-to-face clinic) or at home where the participant will collect a capillary blood sample using a validated home self-test kit (TDL TINY) and send in a prepaid envelope to The Doctors Laboratory (London, UK).

## Data management

Confidentiality of participant data shall be observed at all times during the study. Personal details for each participant taking part in the research study and linking them to a unique identification number will be held locally on a study screening log in the Investigator Site File at each of the investigation centres. The study identification number will be used on the case report forms, on all the blood samples that are collected throughout the study and FSL 2 and FSL Pro data submitted to study-specific Libreview database (Libreview.com). Names and full addresses will not be used. Electronic data will be stored on password-protected computers. Only members of the research team and collaborating institutions will have password access to the anonymised electronic data. Paper copies of the data will be stored for 15 years. Direct access to the source data will be provided for monitoring, audits, REC review during and after the study. The fully anonymised data may be shared with third parties (EU or non-EU based) for the purposes of advancing management and treatment of diabetes. Standard procedures agreed by the MCTU, chief investigator (CI) and clinical principal investigators are in place for data review, database cleaning and issuing and resolving data queries.

## Analysis
### Statistical methods

All efficacy and safety analyses will be conducted following the ITT principle in which all randomised participants are analysed in their allocated treatment group whether or not they receive their randomised treatment. All baseline, 12-week and 24-week outcome data will be presented descriptively, both overall and within treatment group, using mean (SD), median (IQR) or frequency (percentage), as appropriate. All statistical tests will use a two-sided significance level of 5% (unless otherwise specified). All CIs will be presented at a level of 95% and will be two sided. All statistical analyses will be performed using Stata IC 15 (StataCorp).

The primary outcome analysis will evaluate between group differences in HbA1c levels at the end of the 24-week treatment period. An ANCOVA model will be used, with 24-week HbA1c as the outcome and trial arm effect as the focus, and with adjustment for baseline HbA1c, and the other baseline variables included in the minimisation allocation algorithm as covariates. If more than 10% HbA1c data are missing at 24 weeks (or a >10% difference between missing data percentages in the two arms) multiple imputation will be used in order to implement a more complete ITT analysis of the substantive ANCOVA model (otherwise this will be performed as a sensitivity analysis, with a complete-case analysis used as the primary analysis). The imputation model will include baseline and 12-week HbA1c, all the baseline variables used in the outcome model and any other recorded variables found to be predictive of missingness in the 24-week outcome in exploratory analyses (via a logistic regression model, with terms included using a 10% significance level). Sensitivity

analyses will be performed to (1) examine robustness of how missing data are handled, (2) examine efficacy of treatment through Complier-Adjusted Causal Effect modelling, (3) examine impact of data collected outside the visit window and (4) examine potential impact of COVID-19 on the primary outcome.

Quantitative secondary outcomes will also be analysed using ANCOVA and binary secondary outcomes will be analysed using logistic regression. In each case, the analysis will be adjusted for the baseline value of the outcome and the adjustment factors used in the model for HbA1C. Harms data will be reported descriptively as frequencies and percentages (%), both overall and by intervention arm.

Subgroup analysis will be performed by the addition of interactions between intervention arm and the subgrouping factor. All subgroup analyses will be performed separately. Planned subgroups are: baseline HbA1c; treatment modality; prior participation in structured education course; age group; education level; hypoglycaemia unawareness; deprivation index quintile; sex; ethnic group.

Further sensitivity and/or subgroup analyses will be performed as appropriate. These together with fuller details of the analyses proposed above, and any additional analyses, will be included in a full Statistical Analysis Plan that will be approved by the Trial Steering Committee (TSC) prior to any analysis of the outcome data.

### Economic analysis

The economic evaluation will determine the difference in costs and outcomes generated by the FSL2 device compared with SMBG. The economic evaluation will be conducted from the perspective of NHS/Personal Social Services (PSS) following standard quality design and reporting criteria.[24]

A within-trial cost–utility analysis will compare differences in total costs and differences in quality of life using QALYs derived from the EQ-5D-5L. QALYs will be calculated by attaching available utility weights to the health states generated from the EQ-5D-5L, using area under the curve methods with an assumption of a linear change between time points, controlling for baseline. Person-level costs will be generated for each person in the FSL2 device and SMBG arms from a combination of trial-based resource use with published unit costs, allowing comparison in terms of costs to NHS and PSS.[25] Costs will be compared between the two groups using non-parametric methods.

Modelling the potential effect of the intervention on costs and outcomes beyond the trial period will provide a better idea of overall impact as the benefits of controlling HbA1c are likely to be seen after the endpoint of the trial. Therefore, we will carry out an economic evaluation informed by modelling to estimate longer-term benefits and costs, from an NHS/PSS perspective.

A commercially available cost-effectiveness model, the IMS Centre for Outcomes Research and Effectiveness

diabetes model V.8.5 (IMS Health, Danbury, Connecticut, USA), will be used for this economic evaluation. The model consists of 15 submodels designed to simulate diabetes-related complications, non-specific mortality and costs over time. It also incorporates the costs and effects of hypoglycaemia, so is particularly well suited to this study. Two major validation papers on the IMS Core Diabetes Model (CDM) have been published to date.[26 27] The IMS CDM has also been used in a UK-based recent health technology assessment of CGM commissioned by National Institute for Health and Care Excellence (NICE).[28]

Incremental cost-effectiveness ratios will be calculated in the event of the intervention having either higher costs and better outcomes or lower costs and worse outcomes (no scenarios of dominance, based on QALYs and trial primary outcome). The base case analysis will estimate the mean costs and QALYs across the treatment arms. The overall impact of uncertainty will be assessed by generating cost-effectiveness planes from bootstrapped resamples, and distributional assumptions about the transition probabilities, costs and utility values will be made. Cost-effectiveness acceptability curves will be constructed to show the probability that the intervention is cost-effective for different willingness to pay (WTP) thresholds. Incremental economic analysis using the IMS CORE model will require the model's time horizon to be set to 80 years, which approximates to a lifetime horizon. All costs and effects will be discounted by 3.5%, as per NICE guidance.[25 29]

### Process evaluation

The process evaluation will be undertaken to explain discrepancies between expected and observed outcomes, to understand how context influences outcomes, and to provide insights to aid implementation. Specifically, we will investigate whether treatment is consistent with the behavioural change theories, which underpin it and contextual factors have affected implementation. Process evaluation will use a pipeline logic model, showing causal links between resources, activities and outcomes, integrating the National Institute for Health Behaviour Change Consortium's approach to treatment fidelity[30] and a modified version of Linnan and Steckler's framework for process evaluation.[31] We will describe context qualitatively and take a mixed methods approach to characterising recruitment, reach, dose delivered/received and fidelity, with triangulation between data sources.[32] Free-text response questionnaires will be completed by intervention designers, health professionals and trial participants, and analyses combined with trial data, including FSL2 device utilisation and SMBG data which will be analysed descriptively (including the use of appropriate graphical representation), within arms where appropriate, will be synthesised and findings triangulated appropriately.

### Study management

#### Trial management group

Trial management group comprising the CI, coinvestigators, trial managers, trial statistician, trial health

economist, monitor and data manager will meet quarterly to discuss the operational aspects of the study.

### Trial steering committee

TSC with an independent chair has been appointed. Independent membership of the TSC includes two clinical members (including the chair), two service users, health economist and statistician; non-independent members are CI and statistician coinvestigator (only Statistician voting). Other members of the study team including the Sponsor are invited to TSC meetings as observers only.

### Independent data monitoring committee

An IDMC comprising a clinical chair, another clinical expert and a statistician has been appointed. The IDMC will be informed of all serious adverse events and any unanticipated adverse device effects/events that occur during the study. The IDMC will review compiled adverse event data at periodic intervals. The IDMC will report to the TSC any safety concerns and recommendations for suspension or early termination of the investigation.

### Study monitoring

A detailed risk assessment was completed by MCTU and approved by the sponsor. The procedures, source data transfer modalities and anticipated frequency for monitoring are documented in the monitoring plan. Copies of the risk assessment and the monitoring plan will be stored in the Trial Master File. The MCTU study monitor will be fully independent of both the sponsor and site principal investigators.

Authorised representatives of sponsor, regulatory authority or an Ethics Committee may perform audits or inspections at the recruiting centres, including source data verification.

Any substantial/non-substantial amendments will be managed by the CTUs following an approved quality standard operating procedure. Favourable outcomes will be conveyed to participating sites for implementation following local R&D.

### Patient and public involvement

During the grant application process the study protocol received input from several patient groups from Manchester, Derby and Cambridge as well as the INPUT patient advocacy group. One of the study investigators also has T1D as do two members of the TSC.

### ETHICS AND DISSEMINATION

The study will be conducted in accordance with the Declaration of Helsinki Ethical Principles for Medical Research involving Human Subjects (October 2000). It was approved by Greater Manchester West Research Ethics Committee on 21/03/2019REC reference 19/NW/0081. All participants will be provided with oral and written information about the trial, including the most common AEs, and the procedures involved in the study before obtaining written informed consent. The study

results will be communicated to trial participants and disseminated in peer-review publications and through conference presentations. The data sharing plan is available in online supplemental appendix 1.

**Author affiliations**
[1] Diabetes Department, University Hospitals of Derby and Burton NHS Foundation Trust, Derby, UK
[2] University of Nottingham Faculty of Medicine and Health Sciences, Nottingham, UK
[3] Wellcome Trust-MRC Institute of Metabolic Science, NIHR Cambridge Biomedicl Research Centre, Cambridge University Hospitals and University of Cambridge, Cambridge, UK
[4] Barnard Health, BHR Limited, Hampshire, UK
[5] Manchester Clinical Trials Unit, Division of Population Health, Health Service Research and Primary Care, University of Manchester, Manchester, UK
[6] Academic Department of Diabetes and Endocrinology, Queen Alexandra Hospital, Cosham, Portsmouth, UK
[7] Manchester Centre for Health Economics, Divison of Population Health, Health Service Research and Primary Care, University of Manchester, Manchester, UK
[8] Northenden Group Practice, Northenden, Manchester, UK
[9] Manchester Diabetes Centre, Manchester Royal Infirmary, Manchester University NHS Foundation Trust, Manchester Academic Health Science Centre, Manchester, UK
[10] Centre for Biostatistics, Division of Population Health, Health Service Research and Primary Care, University of Manchester, Manchester, UK
[11] The Adam Practice, Upton, Poole, UK
[12] Institute of Immunology and Immunotherapy, College of Medical and Dental Sciences, University of Birmingham, Edgbaston, Birmingham, UK
[13] Elsie Bertram Diabetes Centre, Norfolk and Norwich University Hospital NHS Trust, Colney Lane Norwich, UK
[14] The Ipswich Diabetes Centre and Research Unit, East Suffolk and North Essex NHS Foundation Trust, Ipswich, UK
[15] Division of Diabetes, Endocrinology and Gastroenterology, Faculty of Biology, Medicine and Health, University of Manchester, Manchester, UK

**Correction notice** This article has been corrected since it was first published.

**Contributors** LL, EGW and ME conceptualised the study. LL, EGW, ME, IC, PN, SN, RAE, CS and HT contributed to the grant application. MB is the lead clinical trial manager. CS, AK and VPT are responsible for statistical analysis. All authors contributed to protocol development at various stages. LL, EGW, ME, IC, PN, SN, HT, GR, SL, NK, CK provides site oversight and are responsible for study conduct at each site. RAE and GG are responsible for the health economic analysis. KB-K is responsible for the process evaluation. EGW and LL wrote the first draft of the manuscript and all authors reviewed and had the opportunity to comment on the content prior to submission. The corresponding author confirms that all co-authors are ICMJE recommendation compliant for the submission of this manuscript. No professional writers have been engaged for the preparation of this manuscript.

**Funding** This work was supported by Diabetes UK grant number 18/0005836. The device manufacturer played no part in design, conduct or any other aspects of the study. The study devices were paid by the National Health Service (NHS) UK. Work was supported by the NIHR Cambridge Biomedical Research Centre. The University of Cambridge has received salary support for MLE from the National Health Service in the East of England through the Clinical Academic Reserve.

**Competing interests** EGW has received personal fees from Abbott Diabetes Care, Dexcom, Eli Lilly, Insulet, Medtronic, Novo Nordisk, Sanofi Diabetes Care. LL has received personal fees from Abbott Diabetes Care, Dexcom, Insulet, Medtronic, Novo Nordisk, Sanofi Diabetes Care. ME has received personal fees from Abbott Diabetes Care, Eli Lilly, Medtronic, Novo Nordisk, Astra Zeneca, Zucara. SN has received personalised fees from QUIN,Roche. NK has received personal fees from Abbott,Eli Lilly,Novo Nordisk, Astra Zeneca, Napp, Sanofi. PN has acted as a clinical expert for NICE Medtech innovation briefing MIB110 relating to FreeStyle Libre system. GR has received lecture and consultancy fees from Abbott Diabetes UK.

**Patient consent for publication** Not required.

**Provenance and peer review** Not commissioned; externally peer reviewed.

**ORCID iDs**
Emma G Wilmot http://orcid.org/0000-0002-8698-6207
Katharine Barnard-Kelly http://orcid.org/0000-0002-3888-3123
Rachel Ann Elliott http://orcid.org/0000-0002-3650-0168
Gerry Rayman http://orcid.org/0000-0003-3331-7015

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
