## [Reviewer comments · BMJ Open]

ARTICLE DETAILS

TITLE (PROVISIONAL)	Flash-glucose monitoring with the FreeStyle Libre 2 compared to self-monitoring of blood glucose in sub-optimally controlled type 1 diabetes: the FLASH-UK randomised controlled trial protocol
AUTHORS	Wilmot, Emma; Evans, Mark; Barnard-Kelly, Katharine; Burns, M; Cranston, Iain; Elliott, Rachel; Gkountouras, G; Kanumilli, N; Krishan, A; Kotonya, C; Lumley, S; Narendran, P; Neupane, Sankalpa; Rayman, Gerry; Sutton, Christopher; Taxiarchi, V P; Thabit, H; Leelarathna, L

VERSION 1 – REVIEW

REVIEWER	Wheeler, Benjamin University of Otago, Women's and Children's Health
REVIEW RETURNED	11-Mar-2021

GENERAL COMMENTS	Thank you for the opportunity to review this manuscript. It presents a protocol paper for an appropriately powered randomised controlled trial investigating the efficacy of the newly registered freestyle Libre 2 in adults with sub-optimal glycaemic control. This study will provide important new evidence for efficacy of this new isCGM system. As they correctly identify RCT data is currently lacking for this device, and limited for the first generation system. The protocol paper is well written, and this data when available will be welcomed by the diabetes community. I only have minor comments: How HbA1c (the primary outcome) is assessed should be made a little clearer. I wasn't completely clear after reading that section. Is it uniform across all participants and sites? including within and between participants? POC vs lab? (or both - if both within the same participant? This would be an important weakness as values differ between the 2) Why is the biostatistician not blinded for analyses? Analysis plan and discussion of the questionnaires used appears to be missing? (listed in Table 1 but no details). Was it intended for there to be no wrapping up discussion or similar?
--

REVIEWER	Al Dawish, Mohamed Prince Sultan Military Medical City, Endocrine & Diabetes
REVIEW RETURNED	16-Mar-2021

GENERAL COMMENTS	Dear, It seems that your proposed study is a promising and can help people with Type 1 diabetes to encourage them in controlling glycemic issues. My suggestion to omit exclusion criteria #9. Lastly, good luck.
--

REVIEWER	Pustozerov, Evgenii Saint Petersburg State Electrotechnical University
REVIEW RETURNED	17-Mar-2021

GENERAL COMMENTS	The study is well-designed, dedicated to the urgent topic and clearly described. The primary and secondary outcomes are clear. Inclusion and exclusion criteria are explicitly stated. Sample size is sufficient. The advantages of the study include its multi-center setting with 8 participating organizations and detailed plan for social-economy analysis. As there are both face to face and virtual study visits, the authors are advised to evaluate the impact of visit modes on the outcomes. Minor edits: The metrics in the paper should be consistent (for example, in page 11 line 16 line: “mmol/L” and “mM” on the same line). This also applies to spaces before metrics and precision in numbers (another example – page 8 line 7: “415 million adults and 520,000 children”).
--

REVIEWER	Bao, Yuqian Shanghai Jiao Tong University Affiliated Sixth People's Hospital, Department of Endocrinology and Metabolism
REVIEW RETURNED	24-Mar-2021

GENERAL COMMENTS	This study is a 24-week randomized controlled study that focused on the type 1 diabetes patients with sub-optimal glycemic control, which compared FSL2 (intervention) with SMBG (control). The result has some certain clinical guiding significance. However, there are some problems need to be concern: Major: 1. In this study, the authors only choose CV and SDSG to assess glucose fluctuation. CGM-derived parameters such as MAGE, MODD, etc. are also commonly used to reflect glucose fluctuation, and are recommended to be included. 2. It is recommended to include TBR and TAR into analysis, based on the 2019 international consensus (clinical targets for continuous glucose monitoring data interpretation: Recommendations from the international consensus on time in range. Diabetes Care. 2019.). Minor: 1. The secondary outcomes: please add anti-hyperglycemic therapy, that is, whether other anti-hyperglycemic agents have been used except insulin.
--

VERSION 1 – AUTHOR RESPONSE

Reviewer Reports:

Reviewer: 1

Mr. Benjamin Wheeler, University of Otago

Comments to the Author:

Thank you for the opportunity to review this manuscript. It presents a protocol paper for an appropriately powered randomised controlled trial investigating the efficacy of the newly registered freestyle Libre 2 in adults with sub-optimal glycaemic control. This study will provide important new evidence for efficacy of this new isCGM system. As they correctly identify RCT data is currently lacking for this device, and limited for the first generation system. The protocol paper is well written, and this data when available will be welcomed by the diabetes community.

Thank you for this positive feedback.

I only have minor comments:

How HbA1c (the primary outcome) is assessed should be made a little clearer. I wasn't completely clear after reading that section. Is it uniform across all participants and sites? including within and between participants? POC vs lab? (or both - if both within the same participant? This would be an important weakness as values differ between the 2)

Before the pandemic each site performed the HbA1c at the local NHS hospital Laboratory. People with diabetes fall into a high-risk group for adverse outcomes related to Covid-19, as such the option of a remote HbA1c was introduced to remove the need to come to the clinic for the HbA1c blood test. This was a capillary blood sample posted to a central laboratory where it was processed on the Tosoh G8 analyser, a well-established assay which yields comparable results to inter-laboratory quality assurance schemes, with excellent linear correlation to standard laboratory methods ($r=0.999$).

Why is the biostatistician not blinded for analyses?

It is not possible to blind the statistician for all the analyses due to the substantial differences between the data collected for the two trial arms. The research team made the decision to keep the statistician blind until the Statistical Analysis Plan is approved by the Trial Steering Committee; finalisation of this has been delayed due to COVID-19 but will be completed during summer 2021 in advance of completion of data collection. It is our policy not to finalise the SAP until relatively close to the time at which the first unblinded analysis will be performed in order to minimise processes and changes which, given the low-risk nature of the FLASH-UK trial and lack of planned interim analyses or planned presentation of outcome data by intervention arm.

Analysis plan and discussion of the questionnaires used appears to be missing? (listed in Table 1 but no details).

Details of the statistical analysis is described in the paper pages 11 & 12. As above full statistical analysis plan will be finalised in summer 2021 in advance of completion of data collection. Details of the questionnaires were excluded due to word count and this has been now included in page 8.

Was it intended for there to be no wrapping up discussion or similar?

Thank you, this has been added.

Reviewer: 2

Dr. Mohamed Al Dawish, Prince Sultan Military Medical City

Comments to the Author:

It seems that your proposed study is a promising and can help people with Type 1 diabetes to encourage them in controlling glycemic issues.

My suggestion to omit exclusion criteria #9.

Lastly, good luck.

Thank you so much for these comments.

We included exclusion criteria 9 (more than 1 episode of severe hypoglycaemia in the preceding 24 weeks) because these individuals would meet the UK NICE criteria for real time continuous glucose monitoring (rtCGM). Randomised controlled trial data have shown that rtCGM significantly reduces the recurrence of severe hypoglycaemia in these individuals. We felt it would be unethical to include them in the study as this would prohibit access to rtCGM.

Reviewer: 3

Dr. Evgenii Pustozarov, Saint Petersburg State Electrotechnical University

Comments to the Author:

The study is well-designed, dedicated to the urgent topic and clearly described. The primary and secondary outcomes are clear. Inclusion and exclusion criteria are explicitly stated. Sample size is sufficient. The advantages of the study include its multi-center setting with 8 participating organizations and detailed plan for social-economy analysis.

As there are both face to face and virtual study visits, the authors are advised to evaluate the impact of visit modes on the outcomes.

Thank you for this important comment. Although we are collecting the visit mode whether it is face to face or virtual, one of the issues we have is some participants use mixture of visits both virtual and face to face. One option would be to look at data based on the interaction at visit 3, 4 and 5 as these are the visits that start and optimise the study device. However, we faced difficulties into appropriately classifying participants into meaningful groups. For example, should we classify participants as having virtual visits if they had at least one virtual visit or if they had visit 3 virtually and one more, or if they had at least 2 out of 3 visits virtually? Another option could be to consider the timing of virtual visits, as this may also affect the outcome. For example, early -at visit 3- virtually and then face to face visit(s) compared with early face to face visits followed by late -at visit 4 or 5- virtual visit(s). At this scenario though, participants who completed a virtual visit only at visit 4 would be difficult to classify. Due to this lack of certainty of the most appropriate classification of participants, and aiming to avoid any potentially inconclusive analysis, after due discussion we decided not to include this amongst the subgroup analyses.

Minor edits:

The metrics in the paper should be consistent (for example, in page 11 line 16 line: “mmol/L” and “mM” on the same line). This also applies to spaces before metrics and precision in numbers (another example – page 8 line 7: “415 million adults and 520,000 children”).

We apologise for this. We have corrected these.

Reviewer: 4

Prof. Yuqian Bao, Shanghai Jiao Tong University Affiliated Sixth People's Hospital

Comments to the Author:

This study is a 24-week randomized controlled study that focused on the type 1 diabetes patients with sub-optimal glycemic control, which compared FSL2 (intervention) with SMBG (control). The result has some certain clinical guiding significance. However, there are some problems need to be concern:

Major:

1. In this study, the authors only choose CV and SDSG to assess glucose fluctuation. CGM-derived parameters such as MAGE, MODD, etc. are also commonly used to reflect glucose fluctuation, and are recommended to be included.

Thank you for this recommendation. We carefully considered additional metrics of glycaemic variability but decided to go with CV and SD as recommended by the International Consensus on Use of Continuous Glucose Monitoring.

International Consensus on Use of Continuous Glucose Monitoring. Danne et . al.
Diabetes Care 2017 Dec; 40(12): 1631-1640.
<https://doi.org/10.2337/dc17-1600>

In addition SD, MAGE and other variability metrics that are not adjusted for mean glucose are correlated with mean glucose.

Figure from Supplemental File accompanying above paper. (Appendix 5).

In addition, there is a very strong linear correlation of SD and MAGE, which suggests MAGE does not add much information independent of SD. Figure 1 also shows that the CV is not well correlated with the mean glucose or HbA1c. This implies that CV adds more valuable information on glycemic variability that is more independent or less influenced by the mean glucose or HbA1c value than the SD.

Additional information regarding correlation between metrics can be found here:

Rodbard D. New and improved methods to characterize glycemic variability using continuous glucose monitoring. *Diabetes Technol Ther.* 2009;11:551–565.

Finally we plan to make data available for independent researchers after the study allowing other researchers carry out additional analyses.

We thank the reviewer for highlighting this important area.

2. It is recommended to include TBR and TAR into analysis, based on the 2019 international consensus (clinical targets for continuous glucose monitoring data interpretation: Recommendations from the international consensus on time in range. Diabetes Care. 2019.).

Thank you. Yes we are already analysing TBR and TAR. These are listed in Table 1 of the paper.

Time spent below target glucose (<3.9mmol/l) (<70mg/dl), < 3.5 mmol/l (63 mg/dl); < 3.0 mmol/l (54mg/dl); < 2.8 mmol/l (50 mg/dl)

Time spent above target glucose (10.0 mmol/l) (180 mg/dl), > 16.7 mmol/l) (300mg/dl).

Minor:

1. The secondary outcomes: please add anti-hyperglycemic therapy, that is, whether other anti-hyperglycemic agents have been used except insulin.

Thank you. We will provide information about use of any other anti-diabetes therapy under participant characteristics at baseline and any new non-insulin medication added during the study.

VERSION 2 – REVIEW

REVIEWER	Wheeler, Benjamin University of Otago, Women's and Children's Health
REVIEW RETURNED	28-May-2021

GENERAL COMMENTS	It still remains a little unclear to me whether point of care Hba1c testing will/was done during the clinic visits, and if so - is a mix of laboratory and POC happening in the same individual? Or is it all laboratory? if mix between POC and various post/in person lab - this needs to be acknowledged as a limitation of the study - as there is data highlighting variation in the result between these devices - particularly at the intra-individual level - if same technique (or an equivalent lab technique) is used for each participant at all time points - then there is no issue. Other than making this comment on limitation (if needed - it may be there is no intra-individual variation in POC vs lab) then I am completely happy and wish the team all the best with their work.
---

REVIEWER	Bao, Yuqian Shanghai Jiao Tong University Affiliated Sixth People's Hospital, Department of Endocrinology and Metabolism
REVIEW RETURNED	28-May-2021

GENERAL COMMENTS	All the concerns raised have been addressed appropriately.
--

VERSION 2 – AUTHOR RESPONSE

Reviewer Reports:

Reviewer: 1

Mr. Benjamin Wheeler, University of Otago

Comments to the Author:

It still remains a little unclear to me whether point of care Hba1c testing will/was done during the clinic visits, and if so - is a mix of laboratory and POC happening in the same individual? Or is it all laboratory? if mix between POC and various post/in person lab - this needs to be acknowledged as a limitation of the study - as there is data highlighting variation in the result between these devices - particularly at the intra-individual level - if same technique (or an equivalent lab technique) is used for each participant at all time points - then there is no issue.

Other than making this comment on limitation (if needed - it may be there is no intra-individual variation in POC vs lab) then I am completely happy and wish the team all the best with their work.

Thank you. To clarify, the capillary blood sample is not a point of care test. It is a laboratory test, analysed in a laboratory, not a point of care analyser. It uses a validated home self-test kit (TDL TINY™) sent in a pre-paid envelope to The Doctors Laboratory Ltd (London, UK). It is sent to London to be processed on the Tosoh G8 analyser, a well-established assay which yields comparable results to inter-laboratory quality assurance schemes, with excellent linear correlation to standard laboratory methods ($r=0.999$).

Reviewer: 4

Prof. Yuqian Bao, Shanghai Jiao Tong University Affiliated Sixth People's Hospital

Comments to the Author:

All the concerns raised have been addressed appropriately.

Reviewer: 1

Competing interests of Reviewer: none

Reviewer: 4

Competing interests of Reviewer: No